

# Wearing N95 masks decreases the odor discrimination ability of healthcare workers: a self-controlled before-after study

Guanguan Luo[*], Xingnan Zou[*], Xianlong Zhou, Jiaohong Gan, Cheng Jiang, Zhigang Zhao and Yan Zhao

Emergency Department, Zhongnan Hospital of Wuhan University, China, Hubei, Wuhan
[*] These authors contributed equally to this work.

## ABSTRACT

**Objective**. During the coronavirus disease 2019 (COVID-19) pandemic, the N95 mask is an essential piece of protective equipment for healthcare workers. However, the N95 mask may inhibit air exchange and odor penetration. Our study aimed to determine whether the use of N95 masks affects the odor discrimination ability of healthcare workers.

**Methods**. In our study, all the participants were asked to complete three olfactory tests. Each test involved 12 different odors. The participants completed the test while wearing an N95 mask, a surgical mask, and no mask. The score for each olfactory test was documented.

**Results**. The olfactory test score was significantly lower when the participants wore N95 masks than when they did not wear a mask (7 *vs.* 10, $p < 0.01$). The score was also lower when the participants wore N95 masks than surgical masks (7 *vs.* 8, $p < 0.01$).

**Conclusion**. Wearing N95 masks decreases the odor discrimination ability of healthcare workers. Therefore, we suggest that healthcare workers seek other clues when diagnosing disease with a characteristic odor.

Corresponding authors
Zhigang Zhao,
drzhaozhigang@163.com,
15193946@qq.com
Yan Zhao,
doctoryanzhao@whu.edu.cn

## INTRODUCTION

Hundreds of volatile organic compounds (VOCs) are emitted from the human body, and the components of VOCs usually reflect the metabolic condition of an individual (*Shirasu & Touhara, 2011*). By 400 BC, Hippocrates had already recognized the diagnostic usefulness of body odors and had reported several disease-specific odors emanating from urine or sputum (*Fe, 1994*). Usually, healthcare workers can perceive abnormal odors, which can assist them in the diagnosis of many diseases such as acute alcohol overdose, diabetic ketoacidosis, organophosphate and some other poisonings (*Bijl, Bomers & Smulders, 2013*; *Bomers & Smulders, 2015*).

The global coronavirus disease 2019 (COVID-19) pandemic started in 2019 and affected millions of people around the world. Because the transmission of COVID-19 mainly occurs

through respiratory droplets, wearing N95 masks can effectively reduce the possibility of human-to-human transmission (*Lee & Wang, 2011*; *Postuma et al., 2015*; *Doty, 2009*). A previous study suggested that the N95 mask can cause an average reduction of 37% in the air exchange volume (*Lee & Wang, 2011*). However, whether wearing an N95 mask impairs odor discrimination remains unclear. In the present study, we aim to design a self-controlled study to test the hypothesis that wearing an N95 mask decreases odor discrimination ability of healthcare workers in the emergency department (ED) of a large teaching hospital located in Wuhan, China.

## MATERIALS & METHODS

### Ethics

This study was performed in a large hospital in Wuhan between February 9 and 31, 2021. The study protocol was approved by the Ethics Committee of Zhongnan Hospital in Wuhan City, Hubei province (2021019), and each participant signed an informed consent form at the time of recruitment.

### Participant recruitment

Posters were used to recruit participants at a large hospital in Wuhan. All the participants were healthcare workers older than 18 years old and less than 65 years old. The exclusion criteria were as follows: (1) significant intellectual impairment; (2) dependency on cigarettes; (3) a history of nasal or brain trauma; (4) the use of drugs may affect the olfactory sensation; (5) anosmia; and (6) allergy to essence or paraffin wax. Pre-test questionnaires were distributed to each participant to obtain the following information: sex, age and history of rhinitis, nasal trauma or operation, smoking, anosmia, and influenza in the last 2 weeks.

### Outcome measurement

A 12-item odor discrimination ability test box produced by the Jiangsu Kinsenheimer Biotechnology Co., Ltd. (Jiangsu, China) was used to test the participants' olfactory discrimination ability. The product contains 12 wax blocks and answer cards, which can be used to score the olfactory function of the participants and indicate whether their olfactory discrimination ability has decreased. The 12 wax blocks are all white rectangular shape (Fig. 1). The 12 answer sheets are as follows: the main component of wax block 1 is phyllyl acetate, the correct answer is banana, and the other three disturbing choices are garlic, tobacco and chocolate. The main component of wax block 2 is apple ester, the correct answer is apple, the other three interference options are onion, jasmine, wood. The main component of wax block 3 is anisaldehyde, and the correct answer is star anise, the other three interfering options are coffee, fruit, and grass. The main component of wax block number 4 is roselinol, the correct answer is rose, the other three interference options are soy sauce, peanuts, garlic. The main component of wax block 5 is ethyl butyrate, the correct answer is pineapple, the other three interfering options are soy sauce, jasmine, and tobacco. The main component of wax block 6 is citral, the correct answer is lemon, the other three interfering options are smoke, peach, resin/rosin; The main component of wax block 7
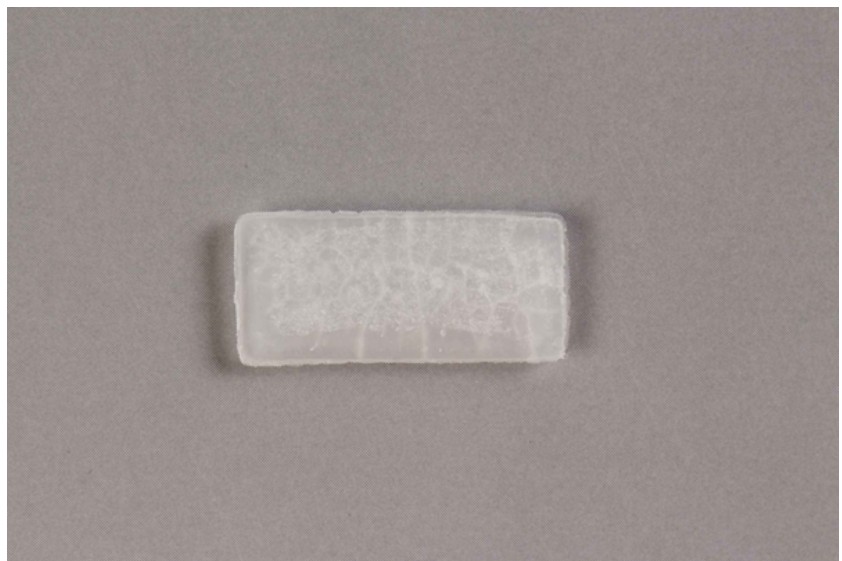

**Figure 1    Experimental wax block.**

is 2, 3-butanedione, the correct answer is milk, the other three disturbing options are strawberry, aniseed/star anise, and smoke. The main component of wax block number 8 is menthol, the correct answer is mint, the other three interfering options are jasmine, rubber tire, and onion. The main component of wax block no. 9 is ethyl silicate, the correct answer is resin/rosin, and the other three interference options are rose, peanut, grass. The main component of wax block 10 is isobornyl acetate, and the correct answer is camphor, the other three interfering options are mint, wood, and soy sauce. For wax block 11, the main component is musk T, the correct answer is wood, the other three interference options are banana, onion, fish. The main ingredient in block 12 is garlic oil, the correct answer is garlic, the other three interfering options are soap, motor oil, and apple. The score was noted as 1 when the participant could correctly identify the odor; in contrast, the score was noted as 0 when the participant could not. The highest possible score is 12, and the lowest score is 0. An olfactory score lower than 8 implies an impairment of an individual's olfactory discrimination ability. This test has been approved to be reliable and mainly used in the diagnostic of Parkinson's disease (*Postuma et al., 2015*; *Doty, 2009*).

## Methods

After completing the pre-test questionnaire, each participant was asked if they had eaten or drunk any food that might affect their sense of smell. They were also asked if they had worn perfume or other odorous cosmetics. After receiving the negative answer, each of them was asked to enter one room that had good ventilation. The 12-item odor discrimination test was administered to each participant when wearing an N95 mask (3M[TM] N95 respirator, catalogue number 1860), surgical mask (Winner[®], Executive Standard: YY 0469-2011) and no mask in turn. Each wax block was presented for approximately 3 s and was held 2–3 cm away from the nostrils. There was an interval of 10 s between each block.

## Sample size calculation

To detect an important difference of 1 in the olfactory test score between the tests performed while wearing an N95 mask and a surgical mask, with a power of 0.9, and type I error of 0.05, the number of participants needed was 57. The sample size was increased to 72 to account for dropouts. In total, 141 participants were asked to complete the pre-test questionnaire. Among them, 3 did not complete the questionnaire. Seventy-one participants were excluded based on the exclusion criteria. It is worth noting that 61 of them were excluded due to dependency on cigarettes. Ten participants were excluded due to nasal trauma and anosmia. Finally, we included 67 participants in our study. The flowchart is shown in Fig. 2.

## Statistical methods

In this study, nonparametric continuous variables were analyzed with the Wilcoxon paired test. The parametric continuous variables were analyzed with Student's $t$-tests. Categorical variables were compared using chi-squared tests. A two-tailed $P$-value $< 0.05$ was considered statistically significant. Statistical analysis of the data was performed with R 4.0.2.

## RESULTS

In this study, 67 participants were finally included. Among them, 37 (55.22%) were male and 30 (44.78%) were female. The mean age of the participants was $31.55 \pm 8.63$ years old. Twelve (17.91%) participants declared that they had a history of rhinitis, while 3 (4.48%) participants reported having had influenza in the last 2 weeks. The data are summarized in Table 1.

Each participant underwent three olfactory tests while wearing an N95 mask, a surgical mask and no mask. Compared to the results obtained when not wearing a mask, the olfactory test score obtained while wearing the N95 mask was significantly lower (10 $vs.$ 7, $p < 0.01$). The score obtained while wearing an N95 mask was significantly lower than that obtained while wearing a surgical mask (7 $vs.$ 8, $p < 0.01$). The data are shown in Table 2 and Fig. 3.

The covariance of the olfactory test score and sex was not significant in the control group that without a mask in Table 3 (9 $vs.$ 10, $p = 0.52$). Meanwhile, we detected the covariance of olfactory test score and age, which was not significant ($\rho = -0.12$, $p = 0.35$). The data are shown in Figs. 4 and 5.

## DISCUSSION

In our study, we found that wearing an N95 mask impaired participants' odor discrimination ability more than wearing a surgical mask. Wearing a surgical mask impaired the odor discrimination ability when compared to not wearing a mask.

Interestingly, different smells have different degrees of recognition. The smell most easily identified by participants wearing N95 or surgical masks was star anise, and the smell most likely to be identified by participants not wearing masks was milk. Of all the participants, with or without masks, only a few could identify camphor, probably because they used it very little, and most people who gave the wrong answer thought it was wood.

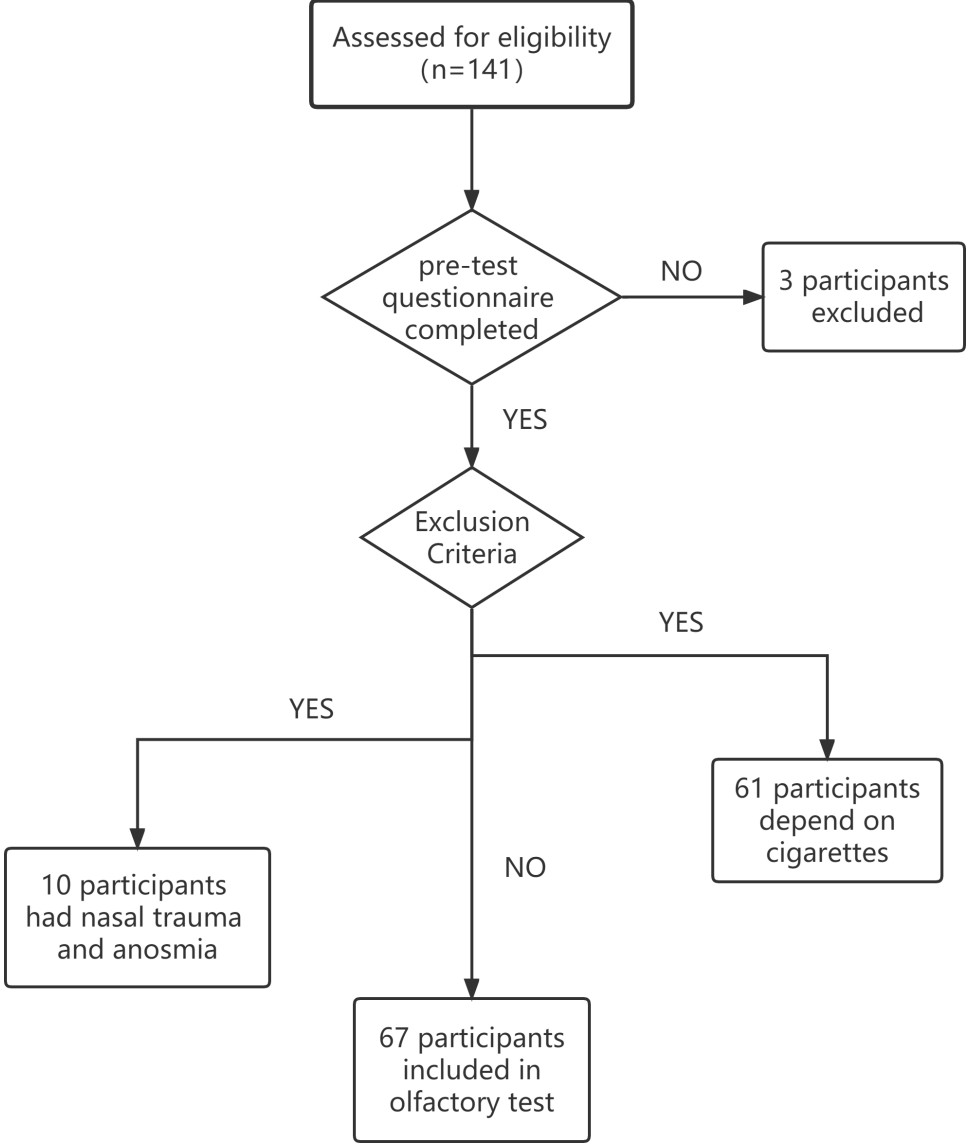

**Figure 2   Flowchart of participant inclusion.**

Sex and age were not correlated with the participants' odor discrimination ability. The number of participants with rhinitis and without rhinitis were significantly different with $p < 0.01$ (Table 1), so we have not compared the olfactory test score of these 2 groups of participants, and the same situation for the 2 groups of having and not having had influenza in the last 2 weeks.

VOCs can be produced by human bodies due to interactions between organic media and biological fluids (*Pavlou & Turner, 2000*). Even in ancient times, before the development of the theory of bacterial pathogenicity, practitioners discovered that the odor of body excretions such as sweat, vaginal fluid, urine and sputum could be changed by different diseases (*Pavlou & Turner, 2000*). Olfactory diagnosis was also a popular method in early

**Table 1 Demographics of the participants.**

| | All participants (n = 67) |
|---|---|
| **Sex, n(%)** | |
| Male | 37 (55.22%) |
| Female | 30 (44.78%) |
| **Age, mean ± SD, years** | 31.55 ± 8.63 |
| **Rhinitis, n (%)** | |
| Yes | 12 (17.91%) |
| No | 55 (82.09%) |
| **Influenza in the past 2 weeks** | |
| Yes | 3 (4.48%) |
| No | 64 (95.52%) |

Notes.
Data was shown in number (percentage).

**Table 2 Olfactory test scores with and without masks.**

| | Median (IQR) | p value |
|---|---|---|
| Without Mask | 10 (8–10) | |
| Surgical Mask | 8 (6.5–9.0) | <0.001 |
| N95 Mask | 7 (6–8) | <0.001 |

Notes.
Data was expressed as median (IQR), wilcoxon paired test was applied in comparison.

traditional Chinese medicine (*Pavlou & Turner, 2000*). In the 1980s, studies showed that the analysis of VOCs could be used to detect certain diseases (*Manolis, 1983*). As technology developed, gas chromatography-mass spectrometry-olfactometry (GC-MS-O), enabled researchers to identify characteristic odor compounds from various biological samples and search for specific odors emanating from patients. This may allow odors to be used as biomarkers of diseases (*Shirasu et al., 2009*). Diagnosis based on odor is still one of the most reliable methods in bedside medicine (*Pavlou & Turner, 2000*).

Some factors, such as aging, neurodegenerative diseases, head trauma, brain tumor extraction, toxin exposure and infection, can significantly affect olfactory discrimination ability (*Beecher, John & Chehrehasa, 2018*; *Dennis et al., 2015*). The exclusion criteria for this study were based on this fact. Studies have shown that olfactory function is impaired in >50% of individuals aged between 65 and 80 years and in 62–80% of those >80 years of age (*Attems, Walker & Jellinger, 2015*). The age range in our study was from 21 years to 60 years; therefore, it is unsurprising that the olfactory test score was not significantly correlated with age. Smoking is also an important factor associated with olfactory dysfunction. According to a systematic review and meta-analysis, current smoking, but not former smoking, is associated with a significantly increased risk of olfactory dysfunction (*Ajmani et al., 2017*). In some studies, male sex was recognized as being associated with reduced olfactory discrimination ability (*Ajmani et al., 2017*). A greater proportion of participants dependent on cigarettes were male (*Bottorff et al., 2014*; *Bolego, Poli & Paoletti, 2002*). To avoid the

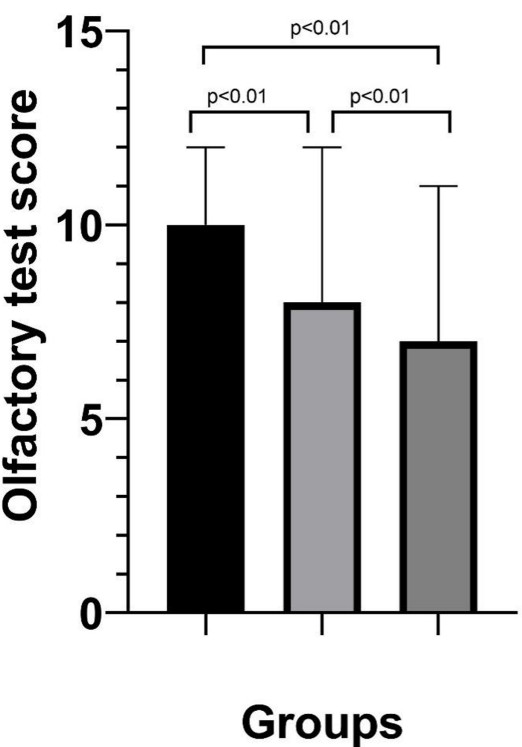
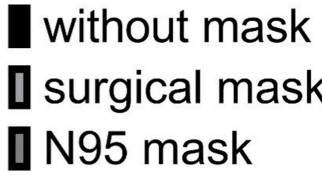

**Figure 3** **Olfactory test scores with and without masks.**

**Table 3** **The effect of sex on olfactory score.**

|  | Median (IQR) | *p* value |
|---|---|---|
| Male | 9 (8–10) |  |
| Female | 10 (8–10) | 0.52 |

Notes.

    Data was expressed as median (IQR), wilcoxon paired test was applied in comparison.

bias caused by smoking, we excluded participants who were dependent on cigarettes and found that sex was not significantly associated with the olfactory test scores in this study.

We designed a fixed order of the experiment: N95 masks, surgical masks and no masks. This order was based on the results of our pilot experiment and the hypothesis that wearing N95 masks would have a relatively more profound effect on odor discrimination. In this study, the very limited number of participants with influenza in past 2 weeks and the number of participants with rhinitis (17.91%) was significantly different from that without rhinitis ($P < 0.01$) lead us to not analyze these two factors. As a result of that, the effects of these 2 factors on the olfactory test score remained unclear and need further study.

According to the recent studies involved the usage of face masks, N95 masks offer considerably better protection from influenza and SARS virus infections when compared to other mask types (*Matuschek et al., 2020*). Depending on the material and dampness, 40–90% of aerosols taking along with the odor molecule were able to penetrate through

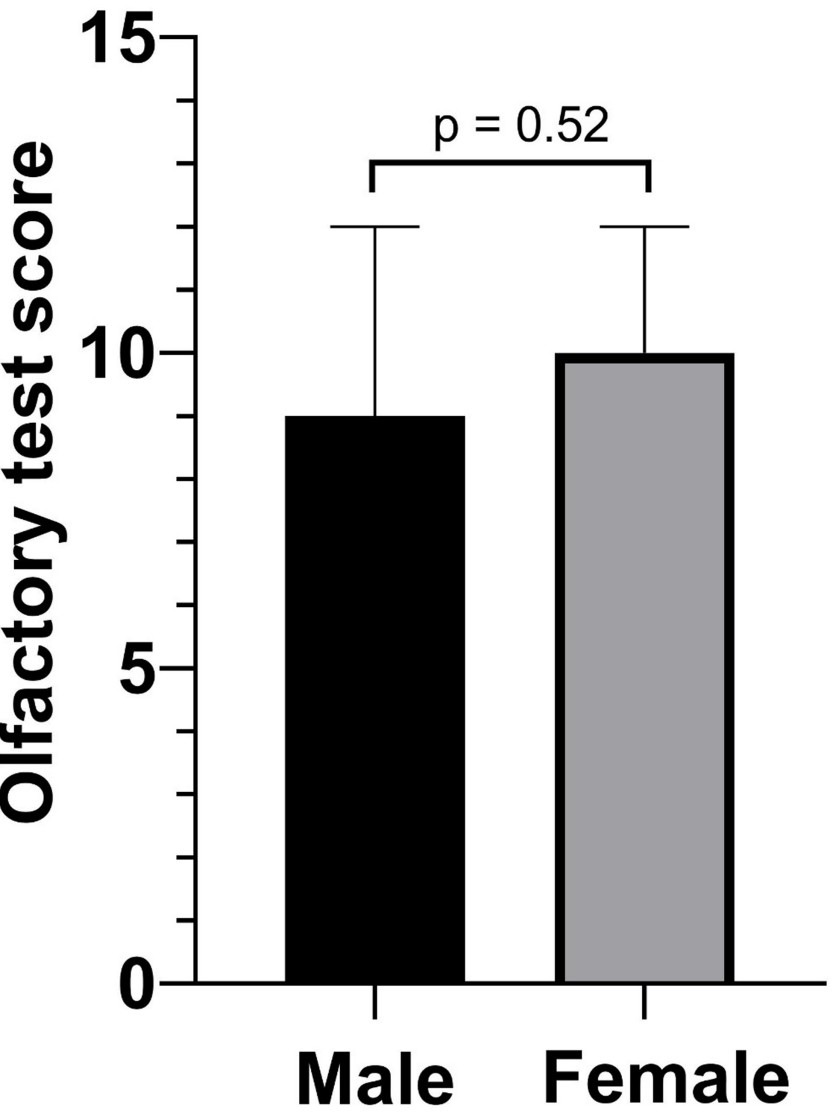

**Figure 4 Olfactory test score according to sex.**

face masks (*Matuschek et al., 2020*). As face masks have become a necessity in the daily work during pandemic of COVID-19, the mechanism of the olfactory effects of the masks still remains unclear and demands for further study.

## CONCLUSIONS

Wearing N95 masks or surgical masks decreased the odor discrimination ability of healthcare workers. Therefore, we suggest that healthcare workers seek other clues to diagnose a disease with a characteristic odor.

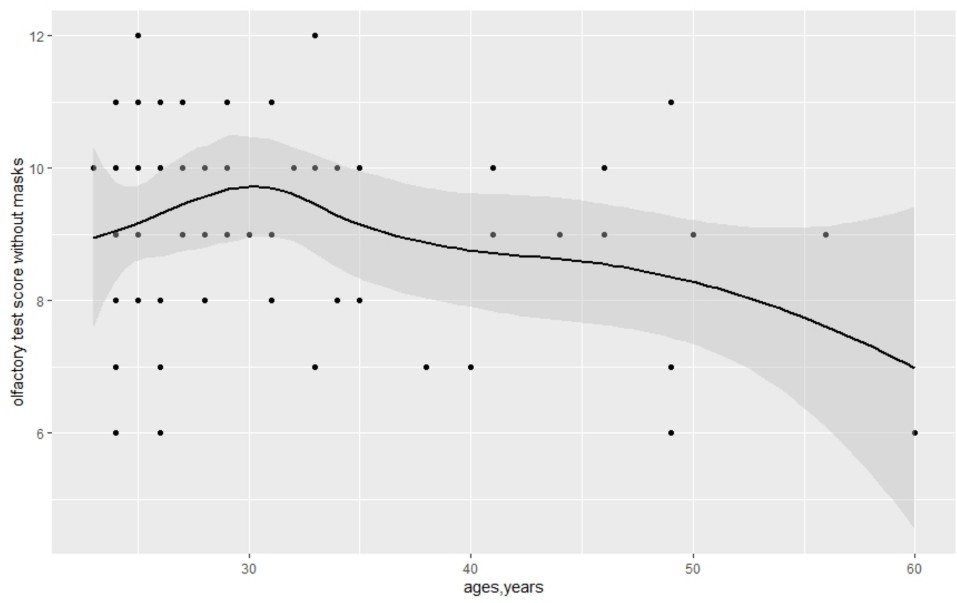

**Figure 5** Covariance of olfactory test scores without masks and age.

## ACKNOWLEDGEMENTS

The authors would like to thank all the participants who made this study possible.

### Funding

This study was supported by the National Natural Science Foundation of China (81900097), the Emergency Response Project of the Hubei Science and Technology Department (2020FCA002, 2020FCA023), and the Emergency Diagnostic and Therapeutic Center of Central China. The funders had no role in study design, data collection and analysis, decision to publish, or preparation of the manuscript.

### Grant Disclosures

The following grant information was disclosed by the authors:
National Natural Science Foundation of China: 81900097.
Emergency Response Project of the Hubei Science and Technology Department: 2020FCA002, 2020FCA023.
Emergency Diagnostic and Therapeutic Center of Central China.

### Competing Interests

The authors declare there are no competing interests.

### Author Contributions

- Guanguan Luo conceived and designed the experiments, performed the experiments, analyzed the data, prepared figures and/or tables, and approved the final draft.
- Xingnan Zou conceived and designed the experiments, analyzed the data, authored or reviewed drafts of the article, and approved the final draft.
- Xianlong Zhou performed the experiments, prepared figures and/or tables, and approved the final draft.
- Jiaohong Gan analyzed the data, authored or reviewed drafts of the article, and approved the final draft.
- Cheng Jiang analyzed the data, prepared figures and/or tables, and approved the final draft.
- Zhigang Zhao analyzed the data, authored or reviewed drafts of the article, and approved the final draft.
- Yan Zhao analyzed the data, authored or reviewed drafts of the article, and approved the final draft.

### Human Ethics

The following information was supplied relating to ethical approvals (i.e., approving body and any reference numbers):

The study was approved by the medical ethics committee of Zhongnan Hospital of Wuhan University (2021019).

### Clinical Trial Ethics

The following information was supplied relating to ethical approvals (i.e., approving body and any reference numbers):

The study was approved by the medical ethics committee of Zhongnan Hospital of Wuhan University.

### Data Availability

The raw data is available in the Supplemental Files.

### Clinical Trial Registration

The following information was supplied regarding Clinical Trial registration:

Zhongnan Hospital of Wuhan University.

### Supplemental Information

Supplemental information for this article can be found online at http://dx.doi.org/10.7717/peerj.14979#supplemental-information.

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
