# Peer review of "Wearing N95 masks decreases the odor discrimination ability of healthcare workers: a self-controlled before-after study"

_PeerJ, doi:10.7717/peerj.14979_

## Round 0.1 · original submission · Major Revisions

With the Reviewers' comments in hand, I recommend a major revision of your manuscript before resubmitting.

In particular, both reviewers asked for greater clarity on some methodological aspects of your study. In particular, among other things, they asked for more details about the odor discrimination test, the pilot study, and some crucial aspects of the experimental procedure. Some p values are provided without further details about the statistical test conducted.

On top of that, both reviewers raised concerns that a substantial part of the sample has a history of rhinitis.

On top of that, please carefully address each of the points raised by the two reviewers.

Reviewer 1 ·

Basic reporting

This paper investigated whether using an N95 mask or a surgical mask will affect the ability of odor discrimination among 67 healthcare workers.

Minor points:
Line 139-141: Add a reference for “Even in ancient times… different diseases.”
Line 174: Based on authors’ results, it should be “Wearing N95 masks or surgical masks decreased the …”.
Figure 1: Low resolution for the flowchart.

Experimental design

The research question was meaningful and well-defined.

Major points:
Line 96 (12-item odor discrimination test): Could authors elaborate on which 12-item odor did you use in the test? and which 4 choices (A-D) did you give to each item? Please provide the details or attach it as a supplemental table?
When will participants know the true answer when they complete the test?

Line 165: What is the pilot experiment here? Are these results published? Please clarify or provide more details here.

Validity of the findings

Major points:
Line 134-135: Authors mentioned the p-value here, but didn't show what test it is or any p-value for rhinitis in Table 1.
Line 163-165:
Influenza in the past 2 weeks was small (4.48%), but there was 17.91% rhinitis in the final dataset. I don’t think 17.91% is “the very limited number”. It may affect the results.
Please explain why these 2 factors were not considered as exclusion criteria.

Minor points:
Line 111: You mentioned “Wilcoxon paired test” here and in Table 2, but “wilcoxon signed-rank test” in Table 3. Please be consistent.

Additional comments

Overall, this paper is brief and neat. No outstanding issue was found in the results.
Minor issues may need to be explained or revised.

·

Basic reporting

A picture of the smell test used and a better bibliography on smell testing would be helpful.
Fig.3 Does the factory test score according to sex refer to with a mask or without?

Experimental design

It is not clear how the test administered to the participants is composed, what odors were tested? Did the participants drink or eat before the test? did they have perfume on their clothes?
Were 12 blocks of wax the same color? could they be visually associated with the perfume they contained?
Failure to analyze data from participants with rhinitis or previous flu represents an important bias.
These patients may be hyposmic and affect test results. I recommend that the authors verify the data of these participants and discuss them in the results.
It would be useful in the results to report which odors the patients identified with and without the mask and to describe it in the discussion.

Validity of the findings

The study confirms a widespread hypothesis in the medical field. N95 mask decreases odor discrimination ability. The results need to be improved.

Additional comments

Study can be improved

---

## Round 0.2 · accepted · Accept

I am glad to inform you that the Reviewer is now happy with your revision, and therefore your manuscript is now suitable for publication.

Best,
MTL

·

Basic reporting

no comment

Experimental design

no comment

Validity of the findings

no comment

Additional comments

The responses of the authors were satisfactory.
The changes made to the manuscript have improved it particularly in the methods and discussion section.
I therefore believe that the manuscript is ready for publication.